# Convergence in voice fundamental frequency during synchronous speech

**Abigail R. Bradshaw**●*, **Carolyn McGettigan**

Department of Speech, Hearing & Phonetic Sciences, University College London, London, United Kingdom

* a.bradshaw@ucl.ac.uk

## Abstract

Joint speech behaviours where speakers produce speech in unison are found in a variety of everyday settings, and have clinical relevance as a temporary fluency-enhancing technique for people who stutter. It is currently unknown whether such synchronisation of speech timing among two speakers is also accompanied by alignment in their vocal characteristics, for example in acoustic measures such as pitch. The current study investigated this by testing whether convergence in voice fundamental frequency (F0) between speakers could be demonstrated during synchronous speech. Sixty participants across two online experiments were audio recorded whilst reading a series of sentences, first on their own, and then in synchrony with another speaker (the accompanist) in a number of between-subject conditions. Experiment 1 demonstrated significant convergence in participants' F0 to a pre-recorded accompanist voice, in the form of both upward (high F0 accompanist condition) and downward (low and extra-low F0 accompanist conditions) changes in F0. Experiment 2 demonstrated that such convergence was not seen during a visual synchronous speech condition, in which participants spoke in synchrony with silent video recordings of the accompanist. An audiovisual condition in which participants were able to both see and hear the accompanist in pre-recorded videos did not result in greater convergence in F0 compared to synchronisation with the pre-recorded voice alone. These findings suggest the need for models of speech motor control to incorporate interactions between self- and other-speech feedback during speech production, and suggest a novel hypothesis for the mechanisms underlying the fluency-enhancing effects of synchronous speech in people who stutter.

## Introduction

Synchronised vocal behaviours are ubiquitous across a variety of everyday settings and can be found in every culture [1]. The production of speech in unison by a group of speakers can be observed in as diverse settings as places of worship, schools, sports stadiums, protest marches, military parades and political rallies. Often such behaviours serve to promote social cohesion and bonding amongst their participants [2]. However, as well as studying joint speech behaviours at the level of the goals and intentions of the collective, we can also consider their impact on lower-level processes within the individual; specifically, for the focus of this paper, how might synchronised speech affect the engagement of different systems for speech motor control?

**Data Availability Statement:** Data files for all experiments reported on in this manuscript are openly available on the Open Science Framework (https://osf.io/rs7gk/ DOI: 10.17605/OSF.IO/RS7GK).

**Funding:** This work was funded by a Research Leadership Award from The Leverhulme Trust (https://www.leverhulme.ac.uk), awarded to C.M. (RL-2016-013). The funders had no role in study design, data collection and analysis, decision to publish, or preparation of the manuscript.

**Competing interests:** The authors have declared that no competing interests exist.

A clue that synchronised speech behaviours may radically change the nature of the mechanisms underlying motor control of speech is found in the observation of their powerful efficacy as a temporary fluency enhancer for people who stutter. The speech of people who stutter is characterised by frequent dysfluencies that interrupt the smooth flow of speech, such as syllable repetitions, prolongations, and blocks (tense pauses in between speech sounds). Strikingly, reading a prepared text in synchrony with another speaker (here termed synchronous speech; also referred to as "choral speech" or "choral reading" in this literature) can temporarily reduce the occurrence of such dysfluent events by 90–100% in most individuals who stutter [3]. Disruption to the processing of sensory feedback from the self-voice has long been theorised to play a role in stuttering [for a recent review, see 4]. However, in synchronous speech, the brain simultaneously receives auditory feedback from the self-voice and the voice of the synchronisation partner (known as the accompanist). It is of interest to consider how this concurrent speech feedback might affect sensorimotor integration processes during speech motor control, in both typically fluent individuals and people who stutter.

It is already known that when engaged in a dialogue, interlocutors show a tendency to align with each other across multiple linguistic levels. For example, speakers may converge in their choice of syntactic structures and lexical wordforms during conversational interactions [5]. As well as this alignment in the content of speech, convergence can also be seen in the voices of the interlocutors; that is, they can start to sound more similar in terms of the way in which they produce speech sounds [for a review, see 6]. This vocal convergence has been demonstrated either by direct measurement of acoustic and phonetic voice parameters [7], such as voice fundamental frequency (F0, the acoustic correlate of pitch), formant frequencies such as F1 and F2 (spectral information in the speech signal which determines perceived vowel identity) and speech rate; or through the use of perceptual judgements by independent listeners in AXB designs [8]. In the latter, an external set of listeners are asked to judge whether a participant's utterance during an interaction (stimulus A) sounds more similar to the model utterance from their interlocutor (stimulus X), compared with that participant's baseline utterance when speaking alone (stimulus B)–if listeners tend to choose A more often than B as the stimulus most similar to X, this suggests that the participant's speech has moved perceptibly toward the interlocutor when speaking with them. This vocal convergence has been studied across a range of interactive contexts, including conversations [9], shared-reading tasks [10], and interactive verbal games [11]. Such behaviour has traditionally been interpreted within the conceptual framework of Communication Accommodation Theory [12], which views such responses as a socially motivated strategy for establishing social affiliation with an interlocutor. Accordingly, there is evidence that social factors can influence the extent to which interlocutors converge or even diverge during interactions. For example, greater vocal convergence has been reported when speakers perceive their partner as more attractive [13], or of a higher social status than themselves [14]; conversely, greater divergence was reported when speakers interacted with an insulting interviewer [15].

An alternative theoretical framework for understanding vocal convergence effects is the Interactive Alignment model [5, 16]. This views such convergence as a subconscious process, driven by an automatic priming mechanism. The latest version of the theory [16] proposes that while listening to the speech of an interlocutor, a listener engages in covert imitation of that speech; the listener then uses a 'simulation route' to derive forward models that predict their partner's upcoming speech utterances based on experience of their own speech actions. This covert imitation tends to also co-activate representations within the production system, which can then spill over into the speaker's own productions resulting in overt imitation. This automatic priming is proposed to occur across all levels of linguistic representation, including semantics, syntax and phonology. The resulting convergence in the speech of the interlocutors is thought to then facilitate mutual understanding between them.

In support of this view of convergence as a more low-level and unconscious process, vocal convergence effects have been reported across a range of 'non-social' contexts, such as vowel repetition or speech shadowing tasks, in which participants listen to words produced by a model speaker and are asked to produce each word they hear as quickly as possible [8, 17–20]. Sato et al., [18] asked participants to produce vowel sounds, first cued by orthographic targets, then acoustic targets, and finally by orthographic targets again. Participants were found to show convergence in F0 and F1 towards the acoustic targets, which was sustained during the second presentation of the orthographic targets. These 'after-effects' were interpreted as evidence of offline recalibration of the sensory-motor targets that drive speech production. The authors proposed that these convergence effects should be viewed within the same framework as sensorimotor accounts of speech motor control, in which forward models of the intended or predicted sensory outcomes of speech movements are compared to actual sensory feedback [21–23]. Specifically, they argued that these models should additionally incorporate the influence of external speech inputs on these processes; that is, the external speech environment leads to adaptive changes in the sensory goals for speech, which in turn can result in imitative changes in speech productions.

Further evidence in support of these ideas is found in studies that demonstrated effects of external speech inputs on responses to perturbations of self-produced auditory feedback. Such perturbations can be made in real-time to the F0 or formants of the auditory feedback a speaker receives from their own voice during speech; these typically trigger opposing changes in the speech productions of the speaker in order to compensate for the perceived sensory error [24, 25]. The extent of such speech motor learning has been shown to be affected by both explicit perceptual training with another voice that aimed to shift perception of a phoneme category boundary [26], and by implicit perceptual learning processes triggered by mere exposure to another voice that cued a participant's own speech production [27]. Therefore, other voices can affect both the vocal characteristics of a speaker, and the responsiveness of their speech motor control system to experimentally induced 'sensory errors' in their speech feedback.

This evidence highlights the need for speech motor control research to move beyond studies in which participants speak on their own, to paradigms that try to capture the true dynamics of the interactive contexts in which speech is typically used in everyday life. This challenge to the prevailing approach in cognitive psychology of studying individual minds in isolation was made more broadly by the 'Joint Action' framework, which argues that a comprehensive understanding of mind and behaviour requires the study of the coordinated actions that are so prevalent in everyday interactions among individuals [28, 29]. In order to extend models of speech motor control to account for sensorimotor processes in such interactive contexts, we need more evidence on how control of an individual's speech is affected by the speech and voices of talkers with whom they are interacting.

It is thus of interest to consider the effects on speech motor control of a synchronous external speech input. Specifically, does speaking in synchrony with an accompanist have any influence on the voice of a speaker, beyond the obvious changes in speech timing? If so, do these changes in the voice of the speaker relate systematically to those of the accompanist voice? In samples of people who stutter, it has been shown that the fluency enhancing effects of synchronous speech rely on the presence of spectral information within the accompanist speech signal. A study by Rami, Kalinowski, Rastatter, Holbert and Allen found that synchronous speech with a speech signal that was low-pass filtered (at 100Hz) so as to remove formants but retain glottal source information (including F0) was not effective at reducing stuttering frequency; conversely, a speech signal low-pass filtered to include the source and only partial spectral cues (either F1 only or F1 and F2) was sufficient to induce fluency during synchronous speech [30]. This suggests that, at least in people who stutter, spectral information in the accompanist voice

has an effect on speech motor control and the sound of the speaker's voice. Other research with samples of typically fluent speakers has reported that synchronous speech reduces variability in the pitch, intonation, amplitude and vowel duration of produced speech [31–34]. However, crucially, it is unknown to what extent these changes in speaking style represent convergence with the acoustic characteristics of the accompanist voice. That is, it is possible that any changes in the acoustic features of the speaker's voice during synchronous speech may simply be a by-product of the process of synchronising the timing of one's speech with any external stimulus, be that another voice or a non-speech stimulus (e.g. a metronome).

Overall, this study aims to bring together these different strands of findings in the literature, in order to provide insight into the influence of external speech inputs on speech motor control. Firstly, as outlined above, we know that synchronous speech can result in changes to the speech of a talker, be that a reduction in dysfluency in people who stutter, or reduced variability in speaking style in typically fluent speakers. However, we currently have no insight into the acoustic specificity of these changes in relation to the acoustic/phonetic properties of the other voice. Secondly, it is known that pairs of talkers tend to converge in speaking style during speech tasks, showing changes in their speech that are specific to the acoustic/phonetic properties of their interlocutor's voice. So far however, this has not been investigated for synchronous speech, and so it is yet unknown whether convergence effects would generalise to this task. Thirdly, it is known that perturbations of simultaneous self-voice feedback induce compensatory and adaptive changes in the acoustics of a talker's speech productions that are specific to the precise perturbation applied. These responses have been interpreted within theoretical frameworks as evidence for the prioritisation of auditory feedback for speech sensorimotor control. However, such studies–and the models they inform—have never incorporated the effects of simultaneous *other-voice* feedback on speech sensorimotor behaviours. It is thus currently unknown whether these responses and underlying processes are specific to self-voice feedback, or whether they might also be employed for responding to other-voice feedback during speech motor control.

In order for us to advance our understanding of these effects, the current study aimed to test whether 1) speakers given simultaneous other-voice feedback during speech–by speaking in synchrony with another talker–show changes in their vocal acoustics, and 2) whether these changes are specific to the external acoustic feedback, rather than being driven by some other aspect of speech synchronisation. In order to do this, experiments are needed which test for acoustic convergence during synchronous speech that is specifically dependent on the auditory properties of the accompanist talker. In Experiment 1, we test this by measuring changes in F0 during synchronous speech, specifically testing for upward changes in F0 during synchronisation with a higher-pitched accompanist, but downward changes in F0 in participants who synchronised with a lower-pitched accompanist. Furthermore, in order to test theoretical frameworks that prioritise auditory feedback for speech motor control, it is important to confirm that these convergent changes are indeed primarily driven by acoustic properties of the other talker's voice, and are not equally generated by other types of input like visual speech information. We test this in Experiment 2, in the inclusion of a visual-only synchronous speech condition (where participants could see but not hear the accompanist), as well as an audiovisual condition (where participants could both see and hear the accompanist). Across both experiments, we predicted that participants would show changes in F0 that were dependent on the F0 of the accompanist voice they experienced (i.e. they should increase to a high F0 accompanist, decrease to a low F0 accompanist, and remain unchanged during visual synchronous speech). Conversely, if changes in F0 are simply a by-product of synchronisation of speech timing with an external stimulus, we would expect uniform changes in F0 across these different conditions.

## General methods

The Gorilla Experiment Builder (www.gorilla.sc) was used to create and host all experiments reported in this paper [35]. The online recruitment platform Prolific (www.prolific.ac) was used for participant recruitment. All participants were compensated for their time by payment of £3.75, administered via Prolific. This study received ethical approval from the local ethics officer at the Department of Speech, Hearing and Phonetic Sciences at University College London (approval no. SHaPS-2019-CM-030). All participants gave informed consent prior to taking part in the study.

Data, analysis scripts and stimuli for each of the two experiments reported here are openly available on the Open Science Framework (https://osf.io/rs7gk/ DOI: 10.17605/OSF.IO/RS7GK).

## General procedure

All experiments began with a headphone screening task, to ascertain that participants were wearing headphones and listening in a quiet environment [36]. This task makes use of a perceptual phenomenon called 'Huggins pitch', an illusory pitch percept which relies on dichotic presentation of stimuli. Specifically, each ear receives the same white noise stimulus, but with a phase shift of 180˚ in a narrow frequency band in one channel. When wearing headphones this results in perception of a faint pitch amongst noise; conversely this percept is not generated when played over loudspeakers. On each trial, participants are asked to detect which of three white noise stimuli contains the hidden tone. Participants who failed to reach criterion performance on this task were not permitted to proceed to the main study (see *Data Exclusion*). For experiment 1 this criterion was set to a score of 6/6; for Experiment 2 this criterion was slightly relaxed to a score of at least 5/6, in an effort to reduce the high level of rejection of participants who reported they were in fact wearing headphones.

The first experimental task for all experiments was a solo reading task, which required participants to read aloud a series of sentences presented on the screen while their speech was audio recorded. This was treated as a 'baseline', in order to measure the participant's F0 when not exposed to another voice/engaging in synchronous speech. This was followed by a synchronous speech task, in which participants were again audio recorded while speaking the same set of sentences, this time in synchrony with another person (the accompanist). The experiment ended with a short debrief, in which participants were asked to rate how well they synchronised with the other voice (from 1-not at all to 5-perfectly), and to report whether they noticed anything about that voice. They were further asked what type of headphones they were using, and to rate how loudly they could hear both the other voice and their own voice on a scale of 1 (very quiet) to 5 (very loud). The data from this debrief questionnaire for each of the experiments can be found in the S1 and S2 Files.

## Audio recording

Due to the online nature of this study, we had a limited amount of control over the recording set-ups of our participants, in terms of the type of device and microphone used, and the background environment (e.g. level of noise). This is a clear limitation of the current work, and reflects the fact that this research was conducted during the global COVID-19 pandemic, in which in-person testing was not possible.

Variability in the technological device and software application used for speech recording can affect the measured acoustic signal, via factors such as the type of compression used, the use of filters by different software programmes and differing sampling rates. A small number of studies have begun to provide evidence on the effects of such variation in remote recording set-ups on measurement of acoustic and phonetic parameters in speech recordings [37, 38]. In

general, these indicate that identification of contrasts within speakers such as vowel arrangements tends to be fairly robust across different remote recording set-ups compared to gold-standard laboratory audio recordings; however, measurement of absolute raw acoustic/phonetic parameters such as frequency and duration measures could be systematically affected by the device or software used, sometimes in vowel-specific ways. This was particularly found to be the case for higher frequencies such as measurement of F2. The current study focused on measurement of F0 in speech recordings, which was reported by Sanker and colleagues to not significantly differ across different remote recording set-ups when averaging across vowels (although there were vowel-specific differences) [37]. Their study also found that variability associated with differences in the software application used for recording (e.g. Zoom, Skype, Facebook messenger) was greater than that associated with variability in the recording device (e.g. whether a Mac or PC was used). Based on the findings and recommendations of their study, we implemented a number of methodological decisions in order to mitigate the potential effects of participants' idiosyncratic recording set-ups.

Firstly, the software application that was used to collect the audio recordings was held constant across participants. Recordings were collected via Gorilla audio recording software, which is powered by the WebRTC (Real Time Communication) API within the browser. This software uses the default settings within the browser it is run on, and does not implement any additional algorithms, compression or functions on the recorded data. Audio recordings were saved as MP3 files. Participants were constrained to the use of a laptop or desktop computer to complete the study; completion using mobile phones or tablets was not allowed. The operating system and browser used by each participant across the three experiments can be found in S1 and S2 Files. Furthermore, most analyses were based on within-participant comparisons; that is, F0 measurements were compared across solo reading and synchronous speech tasks within participants. The effects of the recording set-up would therefore have been uniform across these conditions used in these contrasts e.g. if F0 was overestimated in the recordings, this would be uniform across both solo reading and synchronous speech, and so relative comparisons between these conditions would be unaffected. Additionally, when between-group comparisons were made (e.g. between high F0 and low F0 accompanist conditions), a random effect of participant was included in analyses. All audio recordings were individually checked for each participant; persistent/excessive background noise across recordings resulted in exclusion of that participant's data (see *Data Exclusion*).

### Acoustic analysis

Audio recordings from both experiments were analysed using a custom-made script within the software package Praat [39]. This script first isolated the voiced segments of the acoustic signal (by removing pauses and unvoiced consonants) and then took the median F0 value (in Hz) of each sentence. This was run on each participant's audio recordings from the solo reading and synchronous speech tasks.

## Experiment 1

The aim of Experiment 1 was to investigate whether participants showed convergent changes in the F0 of their produced speech towards the F0 of an accompanist's voice during synchronous speech. In a between-subjects design, participants encountered an accompanist with either an unusually high F0 or an unusually low F0, in order to examine potential convergent changes in F0 in both upward and downward directions. Initially, two conditions were tested corresponding to these high and low F0 conditions. After this data was collected and analysed, a further 'extra-low F0' condition was tested, in which participants synchronised with an

accompanist voice with an even lower F0. This was added after observing the distribution of solo reading baseline F0 in the initial sample tested with our *a priori* designed conditions (see *Results*). For simplicity, we report here on the results of a single data analysis incorporating all three conditions.

## Participants

Twenty female participants (mean age = 28.65 years) took part in the initial high F0 and low F0 conditions, with an equal number of participants (10) in each condition. After data collection and analysis, a further 10 female participants (mean age = 28.2 years) were recruited to take part in an additional extra-low F0 condition. These numbers reflect the final participant samples included in analyses (see *Data Exclusion*). All participants in all three conditions were native speakers of English, with most of the sample being of British nationality (one Canadian and one United States).

Evidence on the effects of speaker gender on variability in the extent of vocal convergence observed is inconclusive [6, 9, 17, 40], but has led many studies to restrict their samples to female speakers only; we therefore similarly opted to only recruit female participants for our experiment so as to have same-sex pairs with our female accompanist voice (see *Stimuli*).

## Design

The solo reading task (our baseline task) consisted of 50 trials, with 50 trial-unique sentences. Participants were instructed to start reading each sentence after a visual 3-2-1 countdown presented on the screen, with an interval of 1 second between each number. Participants had 9 seconds to read the sentence (including time taken by the countdown), followed by an inter-trial-interval of 2 seconds. The synchronous speech task asked participants to read the same sequence of sentences, this time in synchrony with audio recordings of another voice (the accompanist). A trial began with three metronome clicks (with an interstimulus interval of 1 second between clicks) followed by the accompanist voice speaking the sentence; participants were told to use the three-click countdown to help them to start speaking at the same time as the accompanist. Again, participants were given 9 seconds to read the sentence, followed by an inter-trial-interval of 2 seconds. Participants were first given 5 practice trials (with novel sentences) to practice speaking in synchrony with the other voice. After each practice trial, they were instructed to adjust their volume to a level at which they could hear both the other voice and their own voice at a loud and clear volume. Following this practice and volume calibration phase, participants were instructed not to make any further changes to their sound volume. A further 50 trials were then presented, consisting of the same set of sentences as in the solo reading task, presented in an identical order. One of these trials was designated as a vigilance trial; for this trial, when the written sentence appeared onscreen, instead of hearing the other voice read the sentence, they unexpectedly heard the voice ask them to read the last word in the sentence. This vigilance trial was included to check that participants were attending to the audio through the headphones, and not simply reading the sentences without listening to the audio. The audio recordings from this trial were checked for accuracy, but not included in further analyses of F0. This thus resulted in a total of 49 audio recordings of interest for each of the two tasks.

## Stimuli

Audio stimuli in the synchronous speech task consisted of audio recordings of a female speaker of Standard Southern British English reading 49 sentences taken from the Harvard IEEE corpus of sentences [41]. The full set of sentences are given in the S1 Table. These

sentences had an average word length of 8, and are designed to be phonetically balanced. The tokens were recorded using the internal microphone of a MacBook Air and the software programme Audacity [42]. These audio recordings were matched for sound intensity via RMS norming in Praat [39]. A custom-made script in Praat [39] was used to shift the F0 of the voice in these recordings either up or down to create separate stimulus sets for the high, low and extra-low F0 conditions. Stimuli for the high F0 accompanist condition were created by shifting F0 up by 2 semitones; stimuli for the low F0 accompanist condition were created by shifting F0 down by 5 semitones. Both these stimulus sets then underwent F0 norming to the average of the median values of sentences within that set; median F0 values of the sentences were normed to 265Hz for the high condition, and 170Hz for the low condition. To create the extra-low condition stimuli, the F0 of the recordings was first shifted down by 9 semitones. In order to preserve the perceived naturalness of the voice in the recordings, a small adjustment to formant spacing was also made to increase perceived vocal tract length using an open-source Praat script [43]; within this script, the 'vocal tract lengthening factor' parameter was set to 1.1, corresponding to a change of just under 2 semitones. The median F0 of these sentences was then normed to 140Hz. Audio stimuli for the five practice trials of the synchronous speech task (consisting of five additional unique sentences, see S1 Table) and the vigilance trial were recorded by the same speaker, and underwent the same processes of F0 normalisation so as to be in keeping with the F0 values of their respective conditions.

Choice of these F0 targets for the initial high and low F0 conditions was guided by data on the distribution of female voice F0s from the UCLA Speaker Variability Database [44, 45]. For the purposes of this study, we looked at data taken from this speech corpus on audio recordings from 50 female speakers of American-English reading a set of 5 of the Harvard IEEE sentences. These were repeated twice within each of three sessions, giving a total number of 30 tokens per participant. Average F0 values across these tokens were calculated for each participant, and upper and lower cut-offs obtained by taking the values two standard-deviations above and below the group average. This resulted in an upper cut-off of 240Hz and a lower cut-off of 175Hz; our values of 265Hz and 170Hz for the high and low conditions were chosen to be outside of these cut-offs, and therefore were expected to be substantially higher/lower than the baseline F0s of our female participants during solo reading. However, on observing the distribution of average F0 values found in our sample during the solo reading baseline task (see *Results*), we decided to test an additional sample of participants with the extra-low F0 condition outlined above, with an even lower accompanist median F0 of 140Hz.

## Data exclusion

During data collection, various checks on data quality and task performance were conducted to exclude problematic participants, who were then immediately replaced. Firstly, participants who failed the headphone check were not permitted to proceed to the main study. Across all three conditions, a total of 15 participants failed this headphone check and were immediately replaced. Of those who passed the headphone check, a further 13 participants were excluded prior to data analysis, either because they failed the vigilance trial (2 participants), had poor quality audio recordings (e.g. excessive background noise) that affected pitch tracking (6 participants), or because the accompanist voice was audible in their audio recordings (e.g. due to headphones with poor sound insulation) and so interfered with pitch tracking of the participant's voice (5 participants). Again, all of these 13 participants were replaced, so as to achieve the target sample size of 10 participants per condition.

Data from individual trials within a participant were excluded if the participant failed to read the sentence correctly (e.g. made a large speech error or missed the sentence completely)

or if there was excessive background noise on that trial that affected pitch tracking. Further, within each task, trials in which F0 values were more than 3 standard deviations from the mean were excluded. These criteria resulted in exclusion of 3.40% of trials across the whole sample.

## Measures and hypotheses

A measure of F0 change from baseline was calculated for each participant by calculating the difference in semitones between their F0 values during solo reading and those during synchronous speech on a sentence-by-sentence basis, and then averaging across these normed values. This measure thus preserves the sign of these differences, and so indicates whether the change was negative (F0 decreased during synchronous speech) or positive (F0 increased during synchronous speech).

Our central hypothesis across all experiments was that F0 changes during synchronous speech would be specific to the accompanist voice experienced. For this experiment, we therefore predicted that participants in the high F0 accompanist condition should show significant increases in F0 from solo reading to synchronous speech, while participants in the low and extra-low F0 accompanist conditions should show significant decreases in F0. At the group level, we predicted that this expected difference in the direction of F0 changes across the conditions would result in a significant group difference, in which the predicted negative F0 change in the low and extra-low conditions would be significantly lower (i.e. more negative) than the predicted positive F0 change in the high condition.

## Results

### Within-participant analysis of convergence

Convergence patterns shown by individual participants in each of the three conditions are shown in Fig 1; this plots the difference (in semitones) between the accompanist F0 and (i) the participant's average F0 at solo reading and (ii) the participant's average F0 at synchronous speech. To determine if each participant showed significant convergence towards the accompanist voice in F0, a two-sided one-sample t-test was run for each participant to compare their sentence-wise F0 change values (synchronous speech minus solo reading) with zero. These tests were used to categorise participants into convergers (significant increase in F0 at synchronous speech in the high condition; or significant decrease in F0 in the low and extra-low conditions), divergers (significant decrease in F0 at synchronous speech in the high condition; or significant increase in F0 in the low and extra-low conditions) or non-convergers (no significant change in F0). The frequencies in each category across high, low and extra-low conditions are given in Table 1; colour coding in Fig 1 is also used to indicate convergence status. Across the whole sample, 16 participants converged and 14 did not converge (no change or diverged); however, these seem to be somewhat unevenly distributed across the conditions, with slightly fewer convergers in the extra-low F0 condition.

### Group analysis of convergence

To check the comparability of our three groups, we first checked that the average F0 of participants during the solo reading baseline task did not significantly differ between the groups (i.e. before they were exposed to our experimental manipulation in the synchronous speech task). A one-way ANOVA found that average F0 at solo reading baseline did not differ between groups ($F_{(2, 27)} = 0.087$, $p = .917$). Average F0 at solo reading baseline ranged from 163.74Hz to 243.79Hz in the group that went on to experience the high F0 condition ($M = 193.86$,

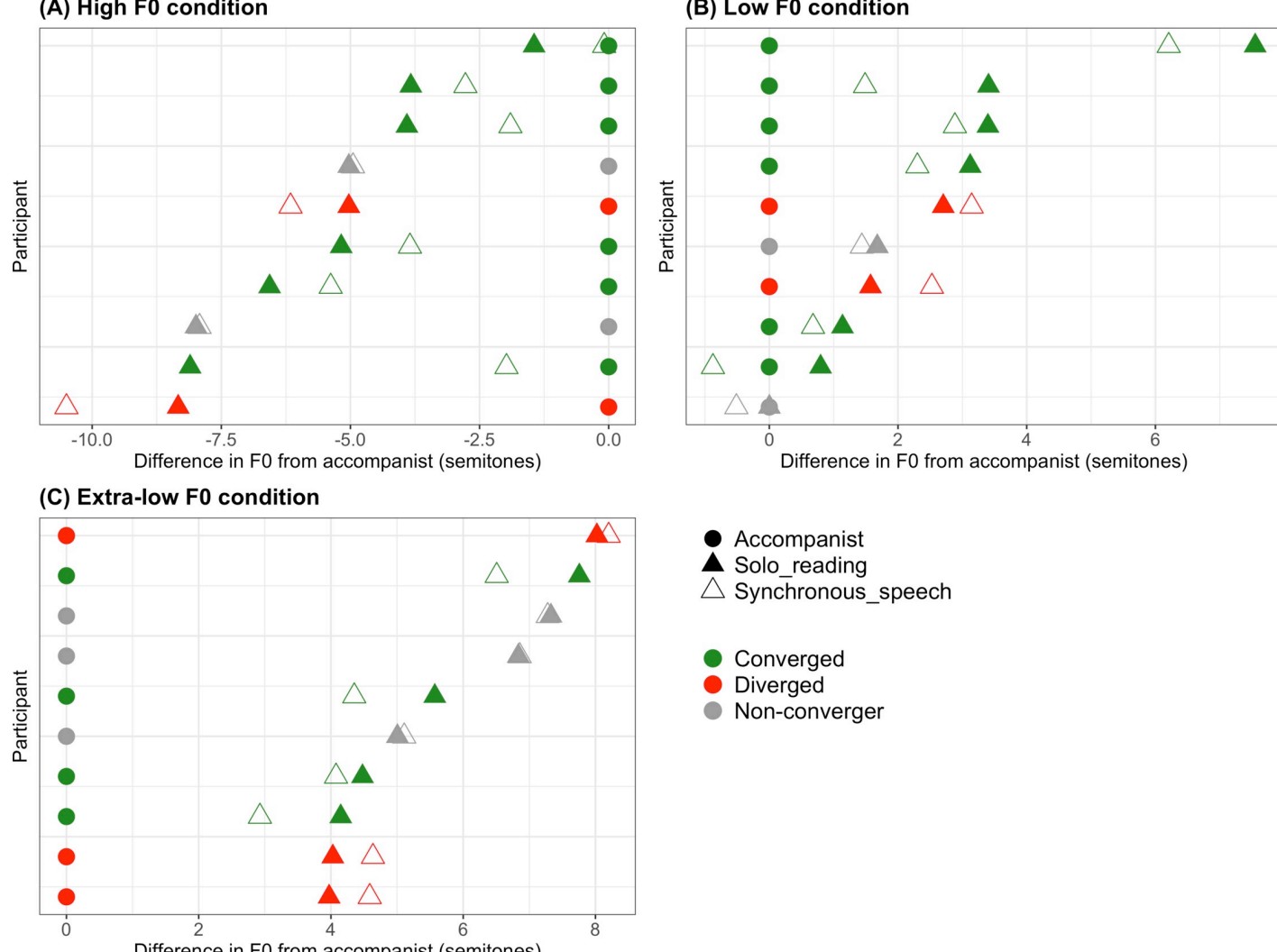

**Fig 1. Individual participant convergence patterns.** Each graph plots for each participant (one per row) the difference in semitones between the accompanist voice mean F0 (represented by the filled circles at zero) and (i) the participant's mean F0 at solo reading (shown in the filled triangles) and (ii) the participant's mean F0 at synchronous speech (shown in the empty triangles). Symbol colour represents whether each participant demonstrated significant convergence, divergence, or no change in their F0 between solo reading and synchronous speech, as determined using one-sample t-tests on their F0 change values (see Table 1). The three conditions (high F0, low F0 and extra-low F0) are plotted on separate graphs (A, B and C).

SD = 25.28) and from 170.01Hz to 262.94Hz in the group that went on to experience the low F0 condition (M = 198.21, SD = 26.07). From these descriptives, it can be seen that the distribution of F0 values measured during solo reading baseline in the low F0 group overlaps with the average F0 of the accompanist voice to which they were subsequently exposed in the

**Table 1. Frequencies of convergers, divergers and non-convergers (no change in F0) in high, low and extra-low F0 groups from Experiment 1.**

| Group | High F0 condition | Low F0 condition | Extra-low F0 condition |
|---|---|---|---|
| Converged | 6 | 6 | 4 |
| Diverged | 2 | 2 | 3 |
| No change | 2 | 2 | 3 |

synchronous speech task (170Hz). Since it was our aim to use accompanist voices whose average F0 was far away from that of our sample at solo reading, this motivated the design of our additional extra-low F0 accompanist condition. In this condition, the accompanist voice's average median F0 (140Hz) was substantially lower than the distribution observed in our initial sample. In the third group tested with this extra-low condition, average F0 at baseline solo reading ranged from 176.11Hz to 222.51Hz ($M = 195.52$, $SD = 18.54$), and thus did not overlap with the average F0 of the accompanist voice they subsequently experienced.

Trial by trial values for the change in F0 from baseline (synchronous speech minus solo reading) are shown for high, low and extra-low conditions in Fig 2; group averages across the whole experiment are shown in Fig 3. It can be seen in Fig 3 that one participant in the high condition demonstrated a noticeably greater change in F0 compared to the group mean (increase of 6.13 semitones); this was within three standard-deviations of the group mean and since no *a priori* criteria were set for outlier detection this participant's data was included in analyses. To test whether the direction of F0 change from solo reading to synchronous speech was significantly different between high and low/extra-low conditions, a linear mixed modelling (LMM) analysis was performed using the lmerTest package in R [46]. A random intercept model was created on F0 change values, with a fixed effect of condition (high, low and extra-low) and random intercepts of sentence and participant. A random slope of condition by sentence was not included due to model convergence issues. This found that F0 change was significantly lower in the low ($\beta = -1.62$, $t(30.01) = -2.67$, $p = .012$) and extra-low conditions ($\beta = -1.25$, $t(30.01) = -2.06$, $p = 0.048$) compared to the high condition (the reference condition in the model; Intercept $\beta = 0.99$, $t(30.14) = 2.32$, $p = .027$). These significant effects reflect the difference in the direction of F0 change between conditions; that is, as predicted, F0 change from solo reading baseline was positive in the high condition, but negative in the low and extra-low conditions. This suggests that the acoustic

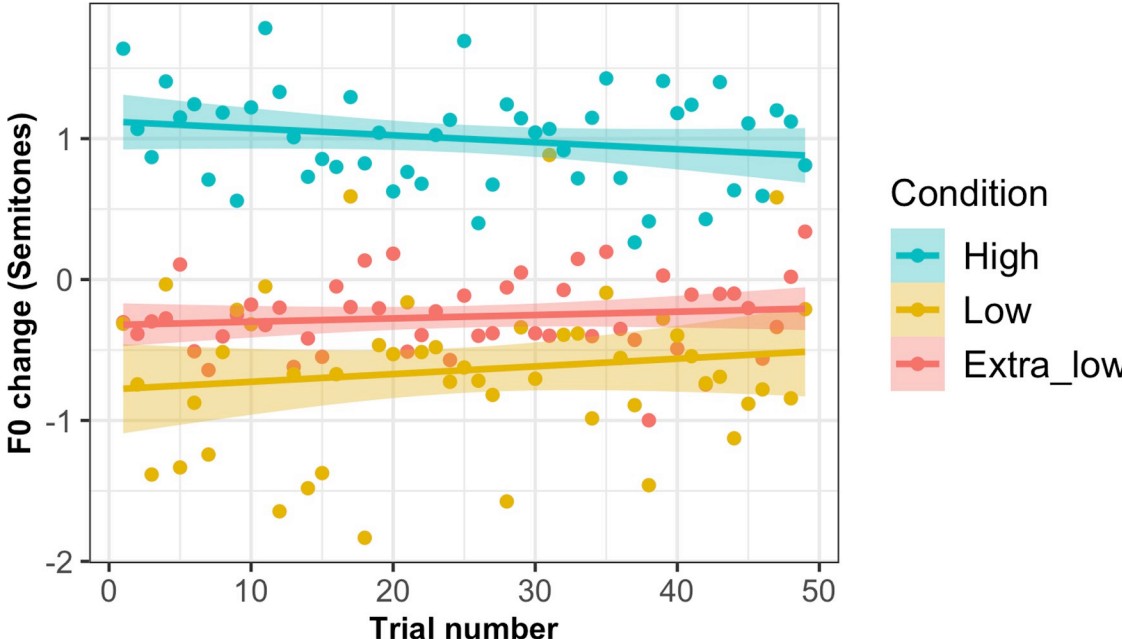

**Fig 2. F0 change across trials in high, low and extra-low conditions.** Graph shows average change in F0 from solo reading baseline to synchronous speech (in semitones) for each trial of the task, averaged across participants in high, low and extra-low conditions. Shaded areas around lines indicate standard error.

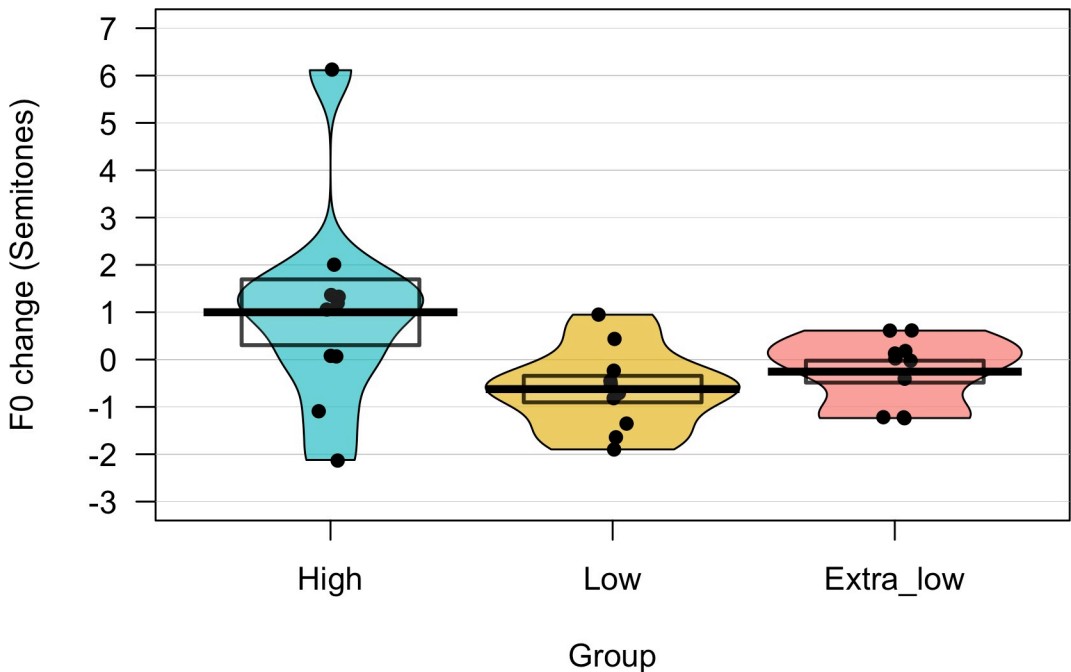

**Fig 3. Average F0 change in high, low and extra-low conditions.** Graph shows average change in F0 from solo reading baseline to synchronous speech (in semitones) averaged across trials for participants in the three conditions of Experiment 1. Dots indicate individual participant averages, thick line indicates group means, boxes indicate standard errors.

properties of the accompanist's voice (specifically, F0) did affect the nature of the change in the participants' voices during synchronous speech.

## Discussion: Experiment 1

Overall, Experiment 1 found evidence of significant convergence in F0 to an accompanist voice during synchronous speech, both when convergence required a raising of F0 (high F0 accompanist condition) and when it required a lowering of F0 (low and extra-low F0 accompanist conditions). That is, changes in F0 observed during synchronous speech were specific to the acoustic properties of the accompanist voice experienced by participants, arguing against the idea that these F0 changes might simply result from the act of synchronisation itself (which would have predicted a uniform increase or decrease in F0 across all conditions).

As previously explained, it was necessary to design and test post-hoc an 'extra-low' F0 accompanist condition in this experiment in addition to the originally planned high and low F0 conditions. This was because the average F0 of our initial low F0 accompanist voice overlapped with the distribution of 'baseline' F0 values within our sample as measured during solo reading. Previous research has suggested that the initial acoustic distance between two speakers can affect the degree of vocal convergence observed, with most studies reporting that convergence is facilitated by a greater distance at baseline [47–49], although the reverse pattern has also been reported [10, 50]. More recently, it has been argued that the former apparent relationship may in fact be an artefact of the way in which convergence is typically calculated, specifically in relation to the use of a 'difference-in-distance' measure (comparing the change in absolute distance between a participant and a model talker from before to after exposure) [51, 52]. In general, Priva and Sanker [51] argued that measurement of convergence can be unreliable when participants' baseline values are close to the model talker, as the influence of

random variability within an individual is likely to overshadow genuine convergent changes, leading to underestimation of convergence or even apparent divergence. This point thus supports our rationale for the inclusion of our extra-low F0 condition, to ensure there was sufficient room for downward convergent changes from solo reading baseline to synchronous speech to be observed. On the other hand, others have suggested that some degree of overlap between the distributions of productions from two interlocutors is necessary for facilitating convergence, so that randomly produced matches between talkers are likely to occur and be reinforced; this was supported by evidence from both model simulations and experimental data [53].

The rationale for including conditions designed to induce both upward and downward convergent shifts in F0 was to determine whether measured changes in F0 during synchronous speech might simply be driven by the act of speech synchronisation itself. That is, it is possible that the act of adjusting the timings of one's speech so as to be in synchrony with an external stimulus could itself induce systematic global changes in F0. There is evidence that speaking conditions associated with increased effort can lead to global changes in F0. For example, when speaking in adverse listening conditions (e.g. background noise or babble), speakers adopt a 'clear speech' speaking style that is accompanied by global changes in F0; Hazan and colleagues [54] reported global increases in mean F0 that did not serve to enhance specific phonological contrasts, suggesting they could reflect a by-product of increased speaking effort. In the field of vocal convergence, a study by Kappes, Baumgaertner, Peschke and Ziegler [55] found reduced convergence in F0 in a speech shadowing task (where participants had to repeat non-words with as minimal a delay as possible, even during the ongoing stimulus presentation) compared to a delayed repetition task; they attributed this to an overall tendency to increase F0 due to increased speaking effort in the shadowing task. In a similar way, synchronous speech could have a global effect on F0 due to the increased effort associated with timing the pace of one's speech with an external stimulus.

Our demonstration in Experiment 1 that F0 changes during synchronous speech appeared specific to the acoustic characteristics of the accompanist voice experienced goes some way to addressing this concern. However, a more direct test of this idea can be achieved through the use of a visual synchronous speech condition. In such a task, speakers are required to produce sentences in synchrony with silent videos of another talker; this thus preserves the synchronisation aspect of the standard synchronous speech condition, while removing the acoustic input from the accompanist voice. We therefore ran a second experiment that compared F0 changes between such a visual synchronous speech condition and an audio-only synchronous speech condition (identical to the high F0 condition of Experiment 1), in order to provide a stricter test of the hypothesis that changes in F0 during synchronous speech depend on the acoustics of the accompanist voice. By testing for significant changes in F0 during visual synchronous speech, this experiment will thus provide greater clarity on the interpretation of the F0 changes observed in Experiment 1. For example, if visual synchronous speech also results in significant increases in F0, these global F0 changes may have contributed to the significant upward shifts observed in the high F0 condition of Experiment 1.

In addition to comparing visual-only and auditory-only synchronous speech conditions, Experiment 2 further tested a group of participants with a combined audiovisual synchronous speech condition. This allowed us to ask whether being able to see as well as hear the person you are synchronising with would affect convergence to the accompanist voice. Previous research has demonstrated that in some circumstances, visual speech information can enhance vocal convergence to auditory speech over listening to that speech alone. Dias and Rosenblum [56] reported that being able to see as well as hear an interlocutor enhanced convergence during a live interactive search task. Conversely, a follow-up study by the same authors found that

visual enhancement of convergence during a non-interactive speech shadowing task was only observed when auditory targets were presented in low-level noise [57]. Thus, enhancement of convergence by audiovisual cues may require either live interaction between speakers, or failing that, challenging auditory conditions. We aimed to investigate this in the context of synchronous speech, by comparing F0 changes across audiovisual and audio-only conditions. Since our speech synchronisation task involves neither live interaction between interlocutors, nor challenging auditory conditions, we might predict based on evidence from Dias and Rosenblum that we would not see enhancement of convergence in our audiovisual condition. Alternatively, synchronising one's speech with a pre-recorded accompanist that can be seen as well as heard may nevertheless increase participants' perception of a social interaction taking place, resulting in enhanced convergence.

## Methods: Experiment 2

### Participants

Thirty female participants (mean age = 27.77 years) took part in this experiment. All participants were native speakers of British English. For this experiment, we only recruited participants who had been born and currently resided in the South East of England, in order to recruit a sample whose accent would match that of the accompanist (see *Stimuli*). An equal number of participants took part in the three synchronous speech conditions (10 in the audio-only condition, 10 in the visual-only condition and 10 in the audiovisual condition). These numbers reflect the final participant samples used in analyses (see *Data Exclusion*).

### Design

Three synchronous speech conditions were tested in this experiment: an audio-only condition in which participants synchronised their speech with a pre-recorded voice (as in Experiment 1); a visual-only condition in which participants synchronised their speech with silent videos of another person speaking; and an audiovisual condition in which participants synchronised their speech with the same videos including access to the audio channel (i.e. the accompanist voice).

Some modifications were necessary for the solo reading and synchronous speech tasks in order to create matched visual-only, audio-only and audiovisual conditions. Primarily, in order to speak sentences in synchrony with videos of a person speaking, the participant cannot be reading a written sentence at the same time as synchronous speech; in the visual-only condition with silent videos this would completely disrupt synchronisation, while in the audiovisual condition this would likely lead to participants synchronising with the audio only and ignoring the video. Instead, visual synchronous speech requires participants to produce sentences that have been previously memorised. In order to achieve this, the number of sentence tokens was reduced down to three sentences from the set used in Experiment 1 (see S1 Table). In the solo reading task, participants produced 15 repetitions of each of these three sentences in a pseudo-random order (in which the same sentence could not appear more than three times in a row), giving a total of 45 trials. For this task, the written sentence was on-screen throughout the trial. All timings were identical to those of the solo reading task in Experiment 1. Participants were instructed to read the sentences after hearing a three-click countdown. In the synchronous speech task, the number and order of sentences to be spoken was identical to the solo reading task; however, this time the sentence was presented onscreen for three seconds before disappearing. The participant then heard the three-click countdown and had 9 seconds to speak the sentence in synchrony with the accompanist. Additionally, after each synchronous speech trial the participant was asked to report whether the sentence they saw/heard the

accompanist speak was the same as the sentence they had been cued to say. This was to test participants' performance on two randomly occurring vigilance trials, in which the accompanist spoke a different sentence to the one the participant had been cued to speak (see *Stimuli*). Participants had up to 5 seconds to report whether there was a mismatch (yes/no) before the next trial began with the presentation of the next sentence onscreen. Participants were given 5 practice trials with these three sentence tokens before completing 45 trials of synchronous speech.

## Stimuli

Stimuli for the three synchronous speech conditions were adapted from three videos of a female speaker of Standard Southern British English. In each video, the speaker read one of three sentences taken from the larger set used in Experiment 1. Stimuli for the visual-only condition were created by removing the audio from these videos. Stimuli for the audio-only condition were created by extracting the audio from these videos, and applying the same pitch shifts as in the high F0 condition of Experiment 1, resulting in an average median F0 of 265Hz for the accompanist. Participants were presented with a fixation cross on the screen for the duration of the audio stimuli. Stimuli for the audiovisual condition were created by recombining the modified audio stimuli from the audio-only condition with the video stimuli. Additionally, vigilance trial stimuli were created for each condition. In these vigilance trial stimuli, the sentence spoken by the accompanist was different to the sentence the participant had been cued to speak. For the visual-only condition, two silent videos of the accompanist speaking two additional sentences selected from the larger set used in Experiment 1 were used as mismatching trials. For the audio-only condition, the corresponding audio from these additional videos were used (again with pitch modifications to norm median pitch to 265Hz). For the audiovisual condition, two types of vigilance stimuli were created: one token in which the video matched the cued sentence but the audio was mismatching; and one token in which the audio matched the cued sentence but the video was mismatching. This allowed us to identify any participants who were relying on one modality only for synchronisation (e.g. synchronising with the voice only while ignoring the concurrent videos). All stimuli in all three conditions began with the same three-click countdown before commencement of speech.

## Data exclusion

Twenty-eight participants failed the headphone check and so were not permitted to proceed to the main study. Of those who passed the headphone check, a further 13 participants were excluded, either due to poor quality audio recordings (5 participants), the audible presence of the accompanist voice in the audio recordings (2 participants), incorrect responses on one or more of the vigilance trials (5 participants), or a failure to follow task instructions (1 participant, who spoke in time with the three metronome clicks in the solo reading task instead of waiting to speak after the countdown). All these participants were replaced, to ensure a final sample size of 30 participants, with 10 participants per synchronous speech condition.

Exclusion of individual trials within a participant's data was again made if the participant made a large speech error or if the F0 value for that trial was more than three standard deviations away from the mean for that participant on that task. These criteria resulted in exclusion of 4.44% of trials across the whole sample.

## Measures and hypotheses

The same measure of F0 change as used in Experiment 1 was calculated for this experiment. Again, our predictions for this experiment stem from our central hypothesis that F0 changes

across conditions will be specific to the accompanist voice experienced. For this experiment, we therefore predicted that participants in the audio-only and audiovisual conditions would show significant increases in F0 from solo reading to synchronous speech (to converge to the high F0 accompanist). Conversely, we predicted that participants in the visual-only condition would show no significant change in F0 between these two tasks. This should therefore result in a significant difference in F0 change between the visual-only condition and the two conditions containing audio. Furthermore, to explore whether convergence to an accompanist voice during speech synchronisation is affected by the addition of visual cues, we also compared F0 change in the audio-only and audiovisual conditions. If being able to see as well as hear the accompanist during synchronous speech has a facilitatory effect on convergence, we would expect F0 change in the audiovisual condition to be significantly greater than that in the audio-only condition.

## Results: Experiment 2

### Within-participant analysis of convergence

Convergence patterns shown by individual participants in each of the three conditions are shown in Fig 4. The frequencies of convergers, divergers and non-convergers across the three synchronous speech conditions are given in Table 2; colour coding in Fig 4 is also used to indicate convergence status. As can be seen, while the majority of participants in the audio-only and audiovisual conditions showed significant convergence, most participants in the visual-only condition showed significant divergence. Average baseline F0 (at solo reading) did not differ between the three groups that went on to experience our three synchronous speech conditions ($F(2,27) = 0.455$, $p = .639$).

### Group analysis of convergence

Trial by trial values for change in F0 from baseline (synchronous speech minus solo reading) are shown for the three synchronous speech conditions in Fig 5; group averages across the whole experiment are shown in Fig 6. To test whether F0 change differed among the conditions, two linear-mixed models were compared: a null model with random intercepts of participant and sentence and a random slope of condition by sentence, and a full model with the same random effects and a fixed effect of condition. A likelihood ratio test found that the full model provided a better fit to the data: $\chi^2(2) = 15.39$, $p < .001$. Pairwise comparisons between conditions using estimated marginal means (with Tukey's HSD adjustment for multiple comparisons) were performed using the emmeans package in R [58]; these found significant differences between the visual-only and audio-only conditions ($t(33.2) = - 2.65$, $p = .032$) and the visual-only and audiovisual conditions ($t(34.3) = -4.19$, $p < .001$). F0 change was thus significantly greater in the two conditions containing audio compared to the visual-only condition, supporting our first hypothesis. Conversely, these did not find a significant difference between the audio-only and audiovisual conditions ($t(34.8) = -1.56$, $p = 0.277$). This is consistent with the hypothesis based on prior literature that the enhancing effects of audiovisual cues on convergence may only be observed during tasks involving live interaction between speakers, or else during challenging auditory conditions.

In addition to comparing F0 change across conditions, it is also of interest to consider whether F0 change in the visual-only condition was significantly different from zero; that is, does synchronising speech with an external stimulus in the absence of any acoustic input lead to any significant changes in F0? To test this, the same model as above was fitted (except with the random slope of condition by sentences removed due to model convergence issues) but this time with the intercept suppressed (set to zero). We then obtained 95% confidence

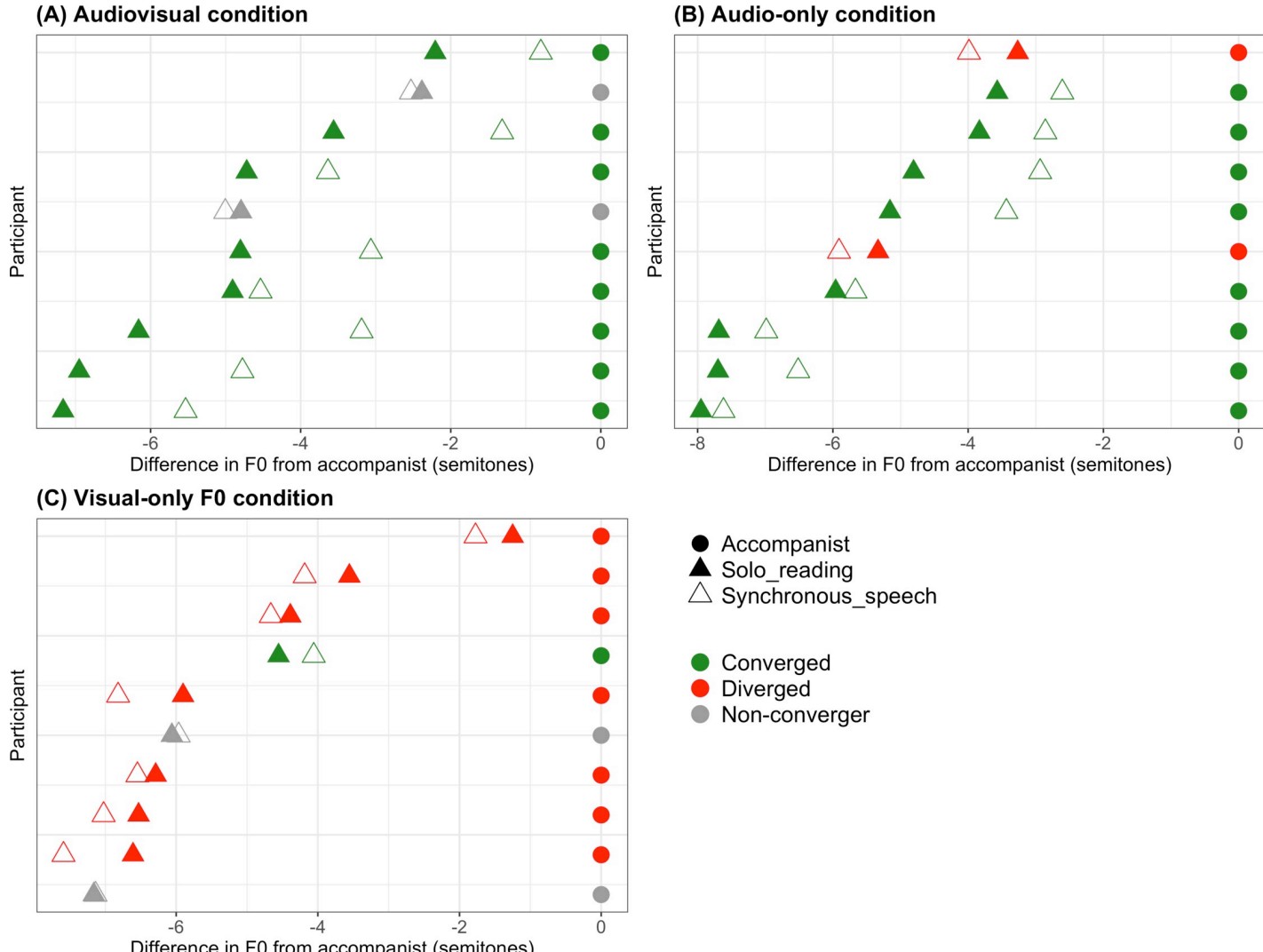

**Fig 4. Individual participant convergence patterns.** Each graph plots for each participant (one per row) the difference in semitones between the accompanist voice mean F0 (represented by the filled circles at zero) and (i) the participant's mean F0 at solo reading (shown in the filled triangles) and (ii) the participant's mean F0 at synchronous speech (shown in the empty triangles). Symbol colour represents whether each participant demonstrated significant convergence, divergence, or no change in their F0 between solo reading and synchronous speech, as determined using one-sample t-tests on their F0 change values (see Table 2). The three conditions (audiovisual, audio-only and visual-only) are plotted on separate graphs (A, B and C).

intervals on the estimates for the three levels of condition using the confint function from the stats package in R [59]. Confidence intervals for the audio-only and audiovisual conditions did not contain zero (all *CIs* [>.15; <1.9]), indicating significant upward changes in F0 from solo reading baseline to synchronous speech. Conversely, confidence intervals for the visual-only

**Table 2. Frequencies of convergers, divergers and non-convergers (no change in F0) in the three synchronous speech conditions from Experiment 2.**

| Group | Visual only condition | Audio only condition | Audiovisual condition |
|---|---|---|---|
| Converged | 1 | 8 | 8 |
| Diverged | 7 | 2 | 0 |
| No change | 2 | 0 | 2 |

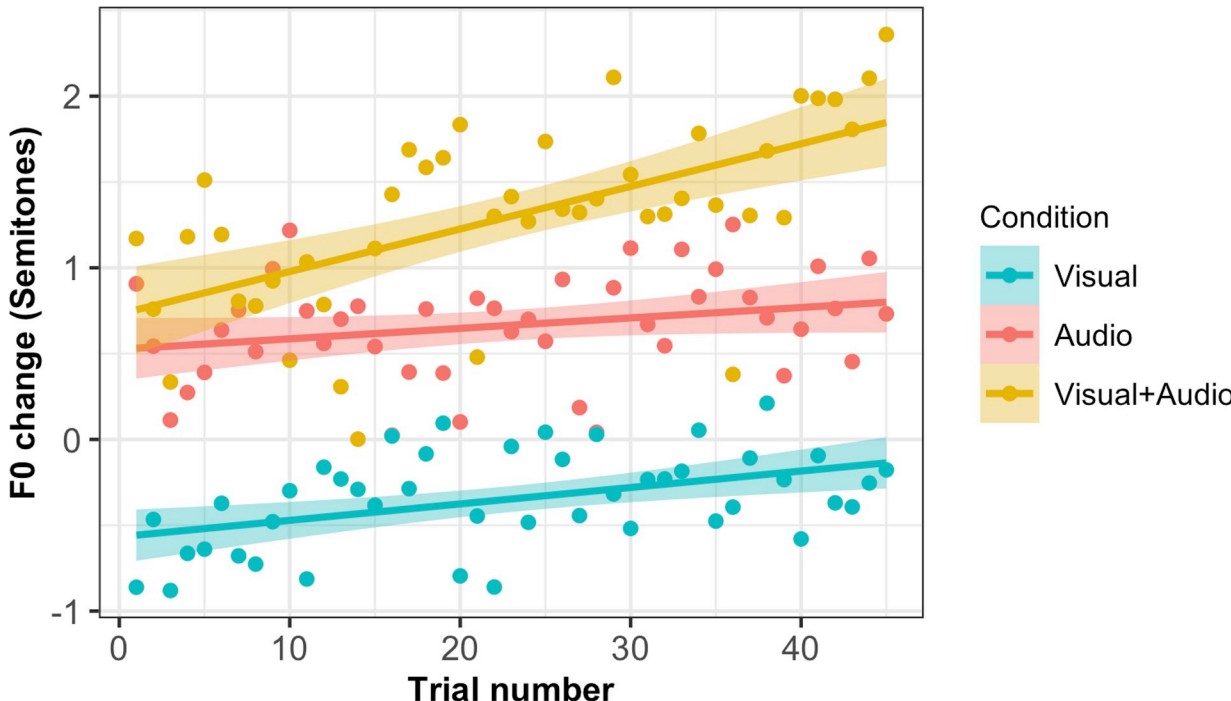

**Fig 5. F0 change across trials in audio, visual and audiovisual conditions.** Graph shows average change in F0 from solo reading baseline to synchronous speech (in semitones) for each trial of the task, averaged across participants in each of the three conditions of Experiment 2. Shaded areas around lines indicate standard error.

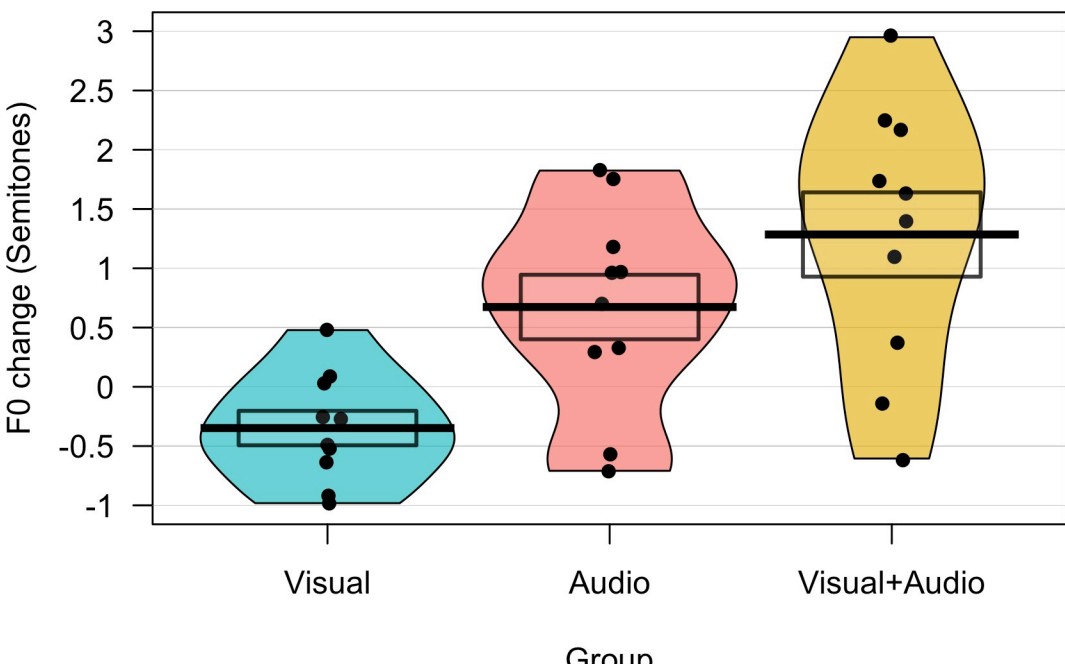

**Fig 6. Average F0 change in visual, audio and audiovisual conditions.** Graph shows average change in F0 from solo reading baseline to synchronous speech (in semitones) averaged across trials for participants in the three conditions of Experiment 2. Dots indicate individual participant averages, thick line indicates group means, boxes indicate standard errors.

condition did contain zero (-0.87; 0.18), indicating that F0 change from solo reading to synchronous speech was not significantly different from zero. This pattern was also seen in the significance of the fixed effects in the zero-intercept model; the effect for the visual-only condition was not significant ($\beta$ = -0.348, $t$(30.36) = -1.35, $p$ = .188), whereas effects for the audio ($\beta$ = 0.675, $t$(30.38) = 2.61, $p$ = .014) and audiovisual conditions ($\beta$ = 1.285, $t$(30.37) = 4.96, $p <$ .001) were both significant.

## Discussion: Experiment 2

In summary, the findings of Experiment 2 first of all replicate our finding from Experiment 1 that synchronisation with a high-pitched accompanist voice induces increases in the F0 of participants' speech productions (audio-only condition). Further, they demonstrate that these convergent F0 changes are not seen when synchronising with a silent accompanist (visual-only condition). Lastly, being able to see as well as hear the accompanist (audiovisual condition) did not have a significant effect on the magnitude of F0 convergence observed, compared to our audio-only condition.

It should be noted that there was some evidence of changes in F0 in the visual-only synchronous speech condition; in contrast to the convergent upward changes in F0 seen in the audio-only and audiovisual conditions to the high F0 voice, a number of participants in the visual-only condition showed a tendency to slightly decrease their voice F0 (demonstrated in the high number of participants showing apparent 'divergence' in Table 2). This could be driven by a slowing of speech rate and production of flatter intonation patterns during synchronisation (which would affect our median measure of F0 across each sentence). These changes in F0 were however not significant at the group level in our zero-intercept model. In Experiment 1, the best convergence appeared to be seen in the high F0 accompanist condition; this demonstration that visual-only synchronous speech does not induce upward changes in F0 therefore suggests that these significant upward shifts are being driven by exposure to the high F0 of the accompanist voice, and thus reflect true convergence.

Interestingly, other work has reported evidence of convergence during speech shadowing of single words in silent visual speech; using an AXB procedure, Miller, Sanchez and Rosenblum [60] found that speakers' productions of single words during lipreading and speech shadowing of a silent model talker were judged as sounding more similar to that model than their non-shadowing utterances. They suggested that visual speech cues can thus provide perceivers with information on the phonetic content and articulatory style of a speaker, that can drive alignment in their speech productions. In contrast, our results suggested a tendency for some of our participants to effectively diverge in F0 from the accompanist during visual synchronous speech, due to a slight decrease in F0 (although these changes in F0 were not significant at the group level). This discrepancy from the previous study may lie in differences in the paradigms used i.e. single word shadowing versus synchronised production of sentences. Alternatively, this difference may lie in our focus on a single acoustic measure (F0); we focused on F0 convergence because this was explicitly manipulated in the accompanist voice, and was considered more likely to be robust when measured from audio recordings collected remotely online compared to formant frequencies [37]. It is entirely possible, however, that participants in our study did converge with the accompanist on other phonetic aspects such as formant frequencies that would be missed by our analysis but picked up in AXB judgements as in Miller et al., [60]. Our findings therefore simply suggest that convergent changes in *F0 specifically* during synchronous speech rely on access to acoustic input from the accompanist's voice, and are not likely explained by the mere act of synchronising the timing of one's speech with another (silent) talker.

Furthermore, in our study, the addition of visual speech cues to auditory speech information did not result in enhanced convergence relative to an audio-only condition. While some previous work has reported enhanced convergence for audiovisual conditions [56], this effect appears to be specific to tasks involving live interaction between interlocutors. Our findings with a non-interactive task involving pre-recorded stimuli instead accord with findings from Rosenblum and Dias, who similarly found no enhancement of convergence for an audiovisual condition over an audio-only condition in a speech shadowing task with pre-recorded stimuli [57]. In that study, an enhancing effect of audiovisual cues on convergence for this non-inter-active speech task was only observed when model talker stimuli were presented in low-level noise. Together with our findings, this suggests that in a non-interactive context, visual speech information is only employed to support convergence when auditory speech information is rendered less informative. Interestingly, this appears to be driven by access to speech-relevant articulatory information, and not simply the social relevance of a face stimulus, with Dias and Rosenblum reporting no enhancement of convergence in low-level noise when the mouth area of the model talker was blurred in their video stimuli. Relating this to the Interactive Align-ment Model [5, 16], access to the articulating mouth may have enabled more accurate or stron-ger covert imitation of the model talker's speech via the simulation route, resulting in enhanced priming of representations in the production system and thus increased imitation.

Conversely, full visual access to a live interlocutor within an interactive speech task has been reported to enhance convergence compared to a condition where the interlocutor was fully occluded from view [56]. It is possible therefore that a similar enhancement of convergence would be seen when engaging in synchronous speech with a live accompanist that could be seen as well as heard. In this context, having visual access to a live accompanist may increase the potential for a two-way exchange of social cues, leading to increased motivation for vocal con-vergence in order to facilitate social bonding with them as an interaction partner. In this way, whether and how visual cues are used to support vocal convergence appears to differ across dif-ferent tasks and contexts, particularly with regards to the presence or not of a live interlocutor.

## General discussion

The experiments reported in this paper provide evidence of vocal convergence (in F0) during synchronous speech in female speakers. Experiment 1 demonstrates that speakers shift the F0 of their produced speech towards that of an accompanist voice during synchronous speech, both when this requires a lowering and a raising of F0. Experiment 2 demonstrates that these convergent F0 changes are not seen when synchronising speech with another speaker in the absence of acoustic input, suggesting that they are indeed driven by perception of the accom-panist voice. For this task, being able to see as well as hear the accompanist did not have a sig-nificant effect on the magnitude of F0 convergence. Overall therefore, as hypothesised, changes in F0 during synchronous speech were specific to the F0 of the accompanist voice. This suggests that these changes reflect true convergence, and are not simply a by-product of a synchronisation of speech timing with an external stimulus.

Synchronous speech is an interesting case for models of speech motor control to consider, which typically place processing of self-generated auditory feedback as central to guiding speech productions [21–23]. Across many of these models, the brain is proposed to compare a prediction of the expected or intended auditory feedback from a speech utterance with the actual auditory feedback it receives; this is important for maintaining stability in speech and for enabling any sensory errors to be corrected online (as demonstrated in altered auditory feedback experiments e.g. [24]). However, in synchronous speech, the brain simultaneously receives auditory feedback from the self-voice and the voice of the accompanist. The evidence

from the current study suggests that this scenario results in sensorimotor processes that drive alignment of speech movements to an external target from the other voice, rather than to an internally defined target or prediction. Some evidence on what this might look like at the neural level was reported by Jasmin et al., [61], who found that synchronous speech with a live (but not a pre-recorded) accompanist resulted in a release from speech-induced suppression in the right anterior temporal lobe (the usual reduction in the auditory cortex response to self-produced speech compared to passive listening to that speech). Such speech-induced suppression is typically interpreted as reflecting a cancellation of the response to the auditory speech signal by subtraction of an auditory prediction of expected/intended feedback for that speech utterance (also known as efference copy or the forward model). Interestingly, this release from suppression was not seen in a condition where the other talker spoke a different (i.e. non-matching) sentence simultaneously with the participant, suggesting it is the synchronous and not the simultaneous nature of the other speech input that drives this enhanced response for synchronous speech. The authors suggested that this release from suppression reflects a blurring of the distinction between self- and other-sensory feedback during speech that results in self-generated speech feedback being processed as if it were externally-generated.

Overall, the current study, along with multiple other lines of evidence, highlights the necessity for models of speech motor control to incorporate influences of external speech inputs on sensorimotor processes during speech (as also argued previously by [18]). In particular, these models will need to consider how the brain can balance and prioritise the competing pressures to maintain stable articulatory targets on the one hand, versus the ability to flexibly change these articulatory targets depending on external speech inputs on the other. The use of predictions during speech is widely assumed, but it is unknown how predictions of self- and other-speech feedback interact during speech production, or further whether synchronous (and not simply overlapping or simultaneous) speech among talkers represents a 'special' case for these predictive processes during speech motor control.

One interesting comparison to make between the sensorimotor adaptation responses seen in altered auditory feedback studies and vocal convergence responses concerns their time-scale. Adaptation responses to altered self-voice feedback typically ramp up over a series of trials [62–64]. Conversely, in our first experiment, F0 change did not appear to linearly increase across the task (see Fig 2). This is consistent with some previous studies of convergence; for example, a study of F0 convergence during a shared reading task by Aubanel and Nguyen found that the maximal level of convergence was achieved at the beginning of the task, then remaining relatively stable across the interaction [10]. However, in our second experiment, convergence did appear to show an upward trend across trials, particularly in the audiovisual condition (see Fig 5). One possible reason for this difference in the pattern of convergence between experiments could lie in the use of a restricted stimulus set in Experiment 2 (only three sentences were repeated across trials, in contrast to the trial-unique sentences used in experiment 1). This is perhaps more similar to the design of altered auditory feedback experiments in which the same small set of stimuli is typically repeated across multiple blocks. Some authors have argued that convergence in different cues can take on multiple different temporal patterns, either continuously increasing across an interaction or dynamically fluctuating according to a speaker's motivational state and intentions [65]. It will be useful for future studies of vocal convergence to consider more closely the factors that might affect the dynamics of the convergence response across time.

The current results also have implications for interpretation of the fluency enhancing effects of synchronous speech in people who stutter. Traditionally, these have been attributed to the provision of an external rhythm which enables a switch from reliance on a faulty basal ganglia-cortical route for internally timed speech to reliance on an intact cerebellar-cortical route for

externally timed speech [66, 67]. The present findings suggest that synchronous speech recruits additional sensorimotor processes that drive imitation-like changes in speech productions in typically fluent speakers. This thus provides a link between synchronous speech and other effective fluency-enhancers used with people who stutter that involve changes in the acoustic/phonetic properties of their speech feedback, either through active processes (e.g. speaking in a foreign accent) or through experimental perturbations of auditory speech feedback. Dysfunction in the processing and forward modelling of self-produced auditory speech feedback has been hypothesised in stuttering [68–70, for a review, see 4]. Indeed, there is recent evidence for an absence of speech-induced suppression effects in people who stutter [71, 72], suggesting faulty auditory prediction and internal modelling. Synchronous speech might circumvent these dysfunctional processes by biasing speech motor control away from a reliance on faulty or unstable internally specified targets for speech and towards imitation of the external targets provided by the accompanist voice.

Alternatively, the beneficial effects of synchronous speech in people who stutter may relate to engagement of an entrainment process [73] that drives changes in the acoustics of speech productions. This interpretation of vocal convergence based on dynamical systems theory [74] was put forward by Pardo [75], who suggested that a pair of interacting talkers can be viewed as an informationally coupled dynamical system. Within such a system, a 'magnet effect' occurs in which externally derived information (i.e. the other speaker's voice) acts as a forcing function on internal dynamics; over time, the more dominant talker pulls the less dominant talker into coordination so that their dynamics (i.e. vocal acoustics) more closely resemble each other, resulting in relative (but not absolute) coordination. This kind of entrainment process may provide stability or an alternative route for speech motor control in people who stutter. Further research on the effects of synchronous speech in people who stutter, both in terms of potential convergence of their speech productions with the accompanist and the nature of neural mechanisms of prediction during synchronous speech, would provide insights into these ideas.

Some limitations of the current study must be acknowledged when interpreting the results. Firstly, relatively small sample sizes were used for our conditions ($n$ = 10 per condition). Although these are in keeping with sample sizes used by others in the field of vocal convergence [e.g. 57, 75, 76], our findings should therefore be viewed as preliminary and in need of further replication. It appears promising, however, that we were able to replicate significant convergence across multiple different synchronous speech conditions in two separate experiments. This study is also limited by the use of an online study design and the resulting variability in the recording set-ups used by different speakers, which could have affected our acoustic measurements of F0 [37]. Although several steps were taken to try to mitigate potential effects of this variability, this study would benefit from replication in a laboratory environment where specialist equipment could be used to record participants' voices in person. This would further allow for a wider range of acoustic and phonetic measures to be examined, such as formant frequencies.

The online study design also limited us to the use of a pre-recorded accompanist voice for the synchronous speech task. As already discussed, this lack of live interaction with the accompanist may have resulted in our failure to observe an enhancing effect of audiovisual cues on convergence [56, 57]. Other studies have reported important differences between synchronous speech conditions involving pre-recorded versus live accompanists. Synchronous speech with a pre-recorded accompanist is more difficult, and leads to reduced success in synchronisation [32, 77]. The decreased variability in vowel productions reported for synchronous speech has also been shown to be less pronounced for synchronisation with a pre-recorded versus a live accompanist [31, 32]. Furthermore, as discussed previously, the release from speech-induced

suppression in the right anterior temporal lobe reported for synchronous speech was specific to a live accompanist condition [61]. From these findings, we might therefore predict that even greater vocal convergence to an accompanist might be seen during live interaction. It should be noted, however, that the fluency enhancing effects of synchronous speech in people who stutter do not appear to rely on a live accompanist [78]. Importantly, the unique aspect of live synchronisation is that both talkers can simultaneously align their speech productions with one another, rather than convergence being a one-way-process. This absence of two-way interaction in our design thus limits the informativeness of this study for models of joint action, instead constraining our interpretation to individualistic representational models. It will be important for future research to follow up these preliminary findings with an in-person study of synchronous speech using live interacting pairs, in order to contribute to our understanding of joint speech production.

## Conclusions

In conclusion, our study demonstrates vocal convergence in F0 during synchronous speech in typically fluent speakers; these changes in F0 were specific to the acoustic properties of the accompanist voice, and so are not simply driven by a coordination of the timing of one's speech with an external stimulus. The findings suggest the need for models of speech motor control to be extended to account for influences of external speech inputs on speech production. Further, they provide novel insights into the potential mechanisms behind the fluency enhancing effects of synchronous speech in people who stutter; specifically, our findings suggest that synchronous speech may induce a shift in the balance with which the speech motor control system weights internally-stored versus externally-generated speech targets for guiding speech productions.

## Supporting information

**S1 File. Debrief data Experiment 1.** Self-report data from debrief questionnaire and data on browser/operating system of participants from Experiment 1.
(CSV)

**S2 File. Debrief data Experiment 2.** Self-report data from debrief questionnaire and data on browser/operating system of participants from Experiment 2.
(CSV)

**S1 Table. Sentence stimuli.** Sentences taken from the Harvard IEEE corpus of sentences used in Experiments 1 and 2.
(PDF)

## Author Contributions

**Conceptualization:** Abigail R. Bradshaw, Carolyn McGettigan.

**Data curation:** Abigail R. Bradshaw.

**Formal analysis:** Abigail R. Bradshaw.

**Funding acquisition:** Carolyn McGettigan.

**Investigation:** Abigail R. Bradshaw.

**Methodology:** Abigail R. Bradshaw, Carolyn McGettigan.

**Project administration:** Abigail R. Bradshaw.

**Writing – original draft:** Abigail R. Bradshaw.

**Writing – review & editing:** Carolyn McGettigan.

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
