## [Decision Letter · Decision Letter 0]

6 Jul 2021

PONE-D-21-16330

Synchronised speech and speech motor control: convergence in voice fundamental frequency during choral speech

PLOS ONE

Dear Dr. Bradshaw,

Thank you for submitting your manuscript to PLOS ONE. After careful consideration, we feel that it has merit but does not fully meet PLOS ONE’s publication criteria as it currently stands. Therefore, we invite you to submit a revised version of the manuscript that addresses the points raised during the review process.

I was fortunate to obtain reviews from three experts. Although two reviewers are the standard at Plos One the fact that this manuscript received three reviews is not an indication that I thought it deserved extra scrutiny but came about because anticipating many reviewer rejections based on past experience, I initially invited a larger number and then simply got lucky that three reviewers agreed to provide a review. I hope this will give you a wide range of helpful suggestions for how to improve the paper.

You will see that the reviewers raise important issues that I encourage you to take on board. In addition, I have read the submission myself and have added some comments of my own, which largely overlap with the reviewers. Below I summarise the crucial points but I urge you to go through the reviews carefully to address and clarify all issues raised.

1.Reviewer 1 suggests streamlining the analyses and I concur with this suggestion. In fact, it seems to me that there is some redundancy that leads to conflicting results, especially in Experiment 3: While the difference between audio-only and audio-visual was not significant for difference scores in the first analysis it was so when both phases were considered separately in the second analysis. This may be due to spurious effects to do with the F0 values in the SR phase. In my view, there should only be one set of analyses for each experiment performed on adjusted (more on this below) differences between SR and CS with full random effect structure (i.e. not just random intercepts of participants and sentences but also random slope of Condition by sentences).

2.The theoretical limitations associated with the use of pre-recorded voices with respect to how informative the study can be for joint action models should be clearly acknowledged.

3.Aim for terminological coherence with the literature in the use of ‘choral speech’ and acknowledge other terms and their definitions.

4.If you streamline the F0 analysis as suggested you may want to consider analysing F0-range. Indeed, far-reaching theoretical conclusions about convergence in general based on just one parameter may not be warranted.

5.I agree with Reviewer 1 that the Hypotheses do not flow directly from the Introduction and need to be better motivated early on. This may then remove the need to restate them for each experiment.

6.The alternative – that F0 convergence is a by-product of trying to converge in speech rate, presumably due to greater effort, should be theoretically better motivated.

7.Clarify baseline comparison between groups prior to experimental manipulation so that the condition label terminology does not become confusing. Also, please aim for consistency of labelling the solo reading phase either as that or as baseline.

8.Clarify what distances you are reporting on. In addition to Reviewer 1’s comments on defining the differences I would caution that absolute differences in Hz obscure the non-linear nature of pitch perception so in my view differences should be reported using a log-transformed measure such as semitones.

9.Introduce clear labels for all factors early on. It took the Reviewers and myself a while to figure out what ‘phase’ refers to – this should have been clearly stated when the design is explained.

10.Consider a direct comparison between XLO (Exp 2), LO (Exp 1) and HI (Exp 1) conditions, essentially merging Experiments 1 and 2 into a joint analysis, while maintaining transparency in reporting the order in which they were carried out.

11.In the figure captions, please state clearly what the abbreviations mean and what the units of the differences are (semitones I would hope in a revised version).

Finally, a small issue I noticed – in line 103, shouldn’t it say ‘if listeners tend to chose B more often…’ ?

I hope that these and the Reviewers’ comments will be helpful for you when preparing a revision of this submission.

We look forward to receiving your revised manuscript.

Kind regards,

Vera Kempe

Academic Editor

PLOS ONE

Journal Requirements:

Reviewers' comments:

Reviewer's Responses to Questions

**Comments to the Author**

1. Is the manuscript technically sound, and do the data support the conclusions?

Reviewer #1: Partly

Reviewer #2: Yes

Reviewer #3: Yes

2. Has the statistical analysis been performed appropriately and rigorously? 

Reviewer #1: No

Reviewer #2: Yes

Reviewer #3: Yes

3. Have the authors made all data underlying the findings in their manuscript fully available?

Reviewer #1: Yes

Reviewer #2: Yes

Reviewer #3: No

4. Is the manuscript presented in an intelligible fashion and written in standard English?

Reviewer #1: Yes

Reviewer #2: Yes

Reviewer #3: Yes

5. Review Comments to the Author

Reviewer #1: See attachment. These extra words are necessary because the badly designed form insists on a minimum character count. Here are some more characters: zzzzzzzzzzzzzzzzzzzzzzzzzzzzzzzzzzzzzzzzzzzzzzzzzzz

Reviewer #2: This is an excellent paper that I am happy to recommend for publication. It provides a thorough and relevant positioning of its original research within the context of previous work on convergence, and in discussion makes connections to possible clinical applications associated with stuttering disorders. Moreover, as a paradigm for effective online data collection, with careful consideration of potential problems, validation methods and norming it has interest beyond its immediate research questions.

Specific Comments (by reviewer question)

1. Is the manuscript technically sound, and do the data support the conclusions?

Yes, with alternative explanations appropriately considered. With 10 participants per condition the contrasts are arguably underpowered in the initial experiment, however, this is mitigated by converging results from the additional experiments.

2. Has the statistical analysis been performed appropriately and rigorously?

Yes. In the LMMs it would be helpful to reorder the levels of the condition where appropriate from default alphabetic ordering such that one level ("high") is consistently baseline for comparison (see below).

3. Have the authors made all data underlying the findings in their manuscript fully available?

Yes, using the OSF platform.

4. Is the manuscript presented in an intelligible fashion and written in standard English?

Very well written and throughly intelligible.

Specific Comments (by line)

211 mention subject compensation, presumably accomlished through the Gorilla platform

455 "Therefore, as predicted, F0 change from baseline was positive in the high condition, but negative in the low condition." While this is clearly correct as visualized in Fig 2, to support this assertion from the LMM results you should also provide the (positive) intercept for the baseline "high" condition.

474 "phase" – I had to go back to see what was meant by "phase" here and suggest including something like "phase (solo vs. choral)" to help other readers with that reminder

592 In the earlier LMM the "high" level is baseline whereas here "extra-low" is used as baseline; similarly Fig 2 shows a high:low ordering while Fig 5 shows low:high. Suggest reordering levels away from the default alphabetic ordering to make this consistent.

804 "with Tukey['s HSD] adjustment"

899 "In contrast, our results suggested a tendency for speakers to diverge in F0 from the accompanist during visual choral speech." How can they diverge if they have no access to that cue? A more plausible reason would seem to be something associated with a repetition effect, which you might consider assessing across trials.

913 This paper is relevant here: Dias, J. W., & Rosenblum, L. D. (2016). Visibility of speech articulation enhances auditory phonetic convergence. Attention, Perception, & Psychophysics, 78(1), 317-333.

917 "perhaps via motor-based speech gesture representations" – the implication here is that F0 can be recovered from facial cues. While there is work suggesting that oral tract constriction gestures can be recovered in this way [e.g. Yehia, H., Rubin, P., & Vatikiotis-Bateson, E. (1998). Quantitative association of vocal-tract and facial behavior. Speech Communication, 26(1-2), 23-43.], I don't see a plausible mechanism for inferring F0.

929 "The provision of a dynamic face stimulus in addition to the voice would have increased the salience of the accompanist as a social agent, potentially increasing participants’ motivation for vocal convergence in order to facilitate social bonding with them as an interaction partner." – except that no exchange of social cues takes place with canned stimuli

976 "recruits additional processes" – this may be so, but should also be discussed in the context of entrainment, which might be providing stability

988 "Further research..." It would be interesting and useful for you to speculate on the possible differences between synchronized speech with a single partner vs. true choral speech as part of a synchronized group.

Reviewer #3: The authors report three experiments in which participants spoke in synchrony with recorded speech. The primary issue was whether participants would adapt spoken f0 when synchronizing with a recording of unusually high or unusually low speech. Across experiments, there was a tendency to converge spoken f0 with the recording; however this tendency was stronger for high than for low pitched recordings. In Experiment 3 participants imitated video recordings as well as video+audio for the high-pitched recordings. This experiment demonstrated that the bias for high-pitch convergence was not simply a byproduct of the joint speech task (vocal f0 went down when synchronizing with video only), but also that there was more convergence for video + audio, suggesting that the results may reflect tendencies based on social interactions.

This was an interesting paper and clearly reported. Here are my main concerns.

(1) would be valuable to correlate should know the difference between baseline f0 (from the initial solo reading task) and f0 for synchronization targets with degree of convergence for individuals.

(2) Sample sizes within each condition were surprisingly small (n=10), and this small sample size was not justified. I know data collection during COVID is difficult, but I think a larger sample of online participants should be possible.

(3) Arguments about applications to stuttering should be limited to discussion because there are no persons who stutter in the experiments. The argument in the discussion could be more clearly described. Why is it beneficial to bring speech motor control away from “internally specified targets and towards external targets”?

(4) The shift of f0 was more extreme for low than for high targets. Do we know that both targets sounded similarly natural? Could lower tendency to converge to low targets reflect the greater loss of naturalness from the more extreme shift?

Minor comments

Lines 413-417

This is a somewhat questionable use of t-test as it assumes independent sampling across trials for a participant.

Lines 430-432

Technically correct, but the lowest participant in the low-voice gropu is for all practical purposes identical in pitch to the low voiced target.

Line 455

Why are there 20 degrees of freedom for t. With n=10 per sample, a t-test on independent samples should be df=18 I would think.

Figure 1

Curious that there does not seem to be a strong trend across trials. Why?

Figure 3 (and similar figures)

State what acronyms mean in the caption.

Table 2

Not sure necessary. Also not clear why 20.4 df for each mean (shouldn’t this be 9 per condition?)

Lines 522-523

What cultural differences predict lower f0 for British-English female speech in contrast to American English? Why is this not an artifact of the small sample?

6. PLOS authors have the option to publish the peer review history of their article (what does this mean?). If published, this will include your full peer review and any attached files.

Reviewer #1: **Yes: **Fred Cummins

Reviewer #2: No

Reviewer #3: No

---

## [Author Response · Author response to Decision Letter 0]

26 Aug 2021

Please see attached file with our response to reviewer and editor comments.

---

## [Decision Letter · Decision Letter 1]

21 Sep 2021

PONE-D-21-16330R1Convergence in voice fundamental frequency during synchronous speechPLOS ONE

Dear Dr. Bradshaw,

Your revision has now been scrutinized by the original three reviewers, and while two reviewers are happy with your revision one reviewer recommends rejection based mainly on a perceived discrepancy between small sample size and stated generalizability of theoretical conclusions. However, I would like to see this paper published, also because the sample size did not figure prominently in the initial round of reviews. However, to make sure that the paper accounts for these potential criticisms, and also to ensure that recent suggestions are incorporated, I am sending it back to you for one final round of very minor reviews. If you decide to submit these reviews (which I very much hope you will) I will not send it out for further review but simply check completion and accept.

For the final version, I recommend the following minor edits:

1. In the General Discussion, provide an acknowledgement of the relatively small sample size and an explanation of how this compares with sample sizes of similar published work. Then please include a statement addressing generalizability given your sample size. The aim of this is to give readers who are not working in this area a sense of where the study sits in this respect.

2. Reviewer 1 argues that in their view ‘synchronized’ speech should be reserved for laboratory speech. While I am not in a position to judge whether such use of terminology is warranted and aligns with a common view, please include a sentence clarifying whether you intend synchronous speech to be reserved for laboratory studies or not, and how this aligns with the literature. Again, my aim is to achieve maximal terminological clarity to allow readers from different areas and theoretical persuasions to fully benefit from reading your work.

3. Reviewer 2 provided a clarification of their earlier statement about entrainment – please check how your interpretation aligns with their conceptualization of entrainment and perhaps add a mention of the stated interpretation in your Discussion.

4. Reviewer 3 points out interesting links to the literature on musical synchronization – I leave it up to you whether you see value in pointing out or even elaborating briefly on these links in the general Discussion.

We look forward to receiving your revised manuscript.

Kind regards,

Vera Kempe

Academic Editor

PLOS ONE

Journal Requirements:

Reviewers' comments:

Reviewer's Responses to Questions

**Comments to the Author**

1. If the authors have adequately addressed your comments raised in a previous round of review and you feel that this manuscript is now acceptable for publication, you may indicate that here to bypass the “Comments to the Author” section, enter your conflict of interest statement in the “Confidential to Editor” section, and submit your "Accept" recommendation.

Reviewer #1: (No Response)

Reviewer #2: All comments have been addressed

Reviewer #3: All comments have been addressed

2. Is the manuscript technically sound, and do the data support the conclusions?

Reviewer #1: Partly

Reviewer #2: Yes

Reviewer #3: Yes

3. Has the statistical analysis been performed appropriately and rigorously? 

Reviewer #1: No

Reviewer #2: Yes

Reviewer #3: Yes

4. Have the authors made all data underlying the findings in their manuscript fully available?

Reviewer #1: Yes

Reviewer #2: Yes

Reviewer #3: Yes

5. Is the manuscript presented in an intelligible fashion and written in standard English?

Reviewer #1: Yes

Reviewer #2: Yes

Reviewer #3: Yes

6. Review Comments to the Author

Reviewer #1: The authors have undertaken a substantial revision and the result is in many ways improved. The notion of baseline, and the introduction of motivated hypotheses are welcome and improve the reading of the text.

However, my judgement is that this is a small experiment performed under highly constrained conditions, that provides some suggestive data that might inform future studies. The data do not, in my opinion, warrant the conceptual elaboration given here, because they are slight, and do not speak clearly. Nor could they, under the constrained circumstances.

The conduct of the experiment was difficult and the Huggins pitch test is a valuable guide for work that might be performed under similar circumstances. It did mean that a lot of participants were excluded. In Experiment 2, for example 41 were excluded, while only 30 made the cut, which is diluted further due to the between subjects design, leading to n=10 in each condition. There is nothing intrinsically wrong with this. The exclusions are warranted, the trials are robustly conducted, and the variables measured are simple: median f0 is a simple measure. But these limitations must, to my mind, mean that this is a pilot, or exploratory study. To represent it as otherwise is to inflate it, and to suggest that it has powers it does not have to inform theory. Drawing conclusions related to the interactive alignment model, or to any neuroscientific interpretation, is simply unwarranted.

The analysis produces an aura of formality over slight data. Everything the data have to be offered can be seen in the first figure for each experiment. The linear modelling is, to my cautious mind, not justified. Such modelling is justified if there is reason to believe that the data might reveal underlying regularities, structures or processes. Median F0 in sentences produced under these conditions with n=10 in each condition is not that kind of data. That is my view and the editor may freely diverge from it.

Finally, and I know I am being a bore about this, but I tried to suggest that work in the area of speech produced simultaneously be described using coherent terms. To that end, I have suggested joint speech as an all-encompassing term, with specializations for speech produced under specific experimental conditions. Choral speech has been the term used by Kanlinowski and colleagues, and that can be taken as a useful marker that work belongs in that camp. I have tried to suggest synchronous speech for laboratory speech such as that of the present study. But the abstract begins by spoiling this attempt. I give up in the present instance.

The relevance of the present work to Kalinowski's agenda is tenuous.

I am unwilling to pronounce on the matter of suitability for publication. The review form insists that I make a choice, so I have to select "reject" but am happy to be overruled. The field is riddled with work that appears important, but masks simple observations conducted in conditions that constrain what can be said beyond the experimental context. Let the editors decide.

Reviewer #2: My comments have been addressed and I endorse publication of this paper.

Concerning the authors' request for clarification about the sense of "entrainment" I intended in a comment on the original form of this sentence: "The present findings suggest that [synchronous] speech recruits additional [sensorimotor] processes that drive imitation-like changes in speech productions in typically fluent speakers." I commented that "this may be so, but should also be discussed in the context of entrainment, which might be providing stability". The point I'd like the authors to consider hinges on the 'mechanism' by which convergence effects are induced. They provide the plausible suggestion that the additional feedback from externally timed speech can drive imitation (convergence) effects, though the lack of auditory cortex suppression facilitating this reported by the Jasmin et al. (2016) paper they cite was crucially found only in live partner interaction which undercuts this argment. But the gradual onset of phonetic convergence, seen most clearly in this submission in the visual+audio condition (Fig 5) and reported elsewhere by Pardo, Babel, and many others, can also be interpreted as a strengthening entrainment process. This arises (following Beek et al. 1992) by way of external information derived from sensory cues. These serve as a forcing function on internal dynamics, leading to adjustments in existing patterns of behavior within their intrinsic range and resulting in relative coordination between two systems (as opposed to absolute coordination between physically coupled systems). Discussion of something along these lines as an alternative explanation would be useful in my opinion.

Beek PJ, Turvey MT & Schmidt RC (1992) Autonomous and nonautonomous dynamics of coordinated rhythmic movements. Ecological Psychology, 4, 65-95.

Reviewer #3: I am satisfied with author responses to my review and the other reviews. I look forward to seeing this paper published. Though the authors need not make changes, there are interesting parallels with the musical synchronization literature that the authors may find interesting (papers by Caroline Palmer and Andrew Chang).

7. PLOS authors have the option to publish the peer review history of their article (what does this mean?). If published, this will include your full peer review and any attached files.

Reviewer #1: **Yes: **Fred Cummins

Reviewer #2: No

Reviewer #3: **Yes: **Peter Pfordresher

---

## [Author Response · Author response to Decision Letter 1]

28 Sep 2021

Please see the attached 'Response to Reviewers' document for our point-by-point responses to the Editor's final comments.

---

## [Editor Report · Decision Letter 2]

5 Oct 2021

Convergence in voice fundamental frequency during synchronous speech

PONE-D-21-16330R2

Dear Dr. Bradshaw,

We’re pleased to inform you that your manuscript has been judged scientifically suitable for publication and will be formally accepted for publication once it meets all outstanding technical requirements.

Kind regards,

Vera Kempe

Academic Editor

PLOS ONE
---

## [Editor Report · Acceptance letter]

11 Oct 2021

PONE-D-21-16330R2 

Convergence in voice fundamental frequency during synchronous speech 

Dear Dr. Bradshaw:

I'm pleased to inform you that your manuscript has been deemed suitable for publication in PLOS ONE. Congratulations! Your manuscript is now with our production department. 

Kind regards, 

on behalf of

Prof Vera Kempe 

Academic Editor

PLOS ONE